# Long term fMRI adaptation depends on adapter response in face-selective cortex

Daphne Stam [1], Yun-An Huang[1], Kristof Vansteelandt[1,2], Stefan Sunaert[3,4], Ron Peeters[3,4], Charlotte Sleurs[5], Leia Vrancken[6], Louise Emsell [1], Rufin Vogels[7], Mathieu Vandenbulcke[1,2] & Jan Van den Stock [1,2 ✉]

Repetition suppression (RS) reflects a neural attenuation during repeated stimulation. We used fMRI and the subsequent memory paradigm to test the predictive coding hypothesis for RS during visual memory processing by investigating the interaction between RS and differences due to memory in category-selective cortex (FFA, pSTS, PPA, and RSC). Fifty-six participants encoded face and house stimuli twice, followed by an immediate and delayed (48 h) recognition memory assessment. Linear Mixed Model analyses with repetition, subsequent recognition performance, and their interaction as fixed effects revealed that absolute RS during encoding interacts with probability of future remembrance in face-selective cortex. This effect was not observed for relative RS, i.e. when controlled for adapter-response. The findings also reveal an association between adapter response and RS, both for short and long term (48h) intervals, after controlling for the mathematical dependence between both measures. These combined findings are challenging for predictive coding models of visual memory and are more compatible with adapter-related and familiarity accounts.

[1] Neuropsychiatry, Leuven Brain Institute, KU Leuven, Leuven, Belgium. [2] Geriatric Psychiatry, University Psychiatric Centre KU Leuven, Leuven, Belgium.
[3] Department of Radiology, University Hospitals Leuven, Leuven, Belgium. [4] Deaprtment of Imaging and Pathology, KU Leuven, Leuven, Belgium.
[5] Department of Pediatrics, University Hospitals Leuven, Leuven, Belgium. [6] Laboratory for Experimental Psychology, KU Leuven, Leuven, Belgium.
[7] Laboratory for Neuro- and Psychophysiology, Leuven Brain Institute, KU Leuven, Leuven, Belgium. ✉email: jan.vandenstock@kuleuven.be

The predictive coding hypothesis (PCH) states that the brain is continuously generating predictions about sensory input based on an internal model of the external environment. These predictions are matched with the sensory input and the resulting prediction errors hierarchically serve to validate the internal model (minimizing free energy), with neural activation positively correlating with prediction errors. Furthermore, it has been postulated that the reduced neural response following repeated presentation of a stimulus, termed repetition suppression (RS)[1] reflects a reduction in prediction error[2] and as such facilitation of behavioural responses (priming), e.g. shorter reaction time during identification[3]. While some studies suggest a quantitative relation between priming and RS[4,5] others suggest that RS may not directly relate to visual priming[6–8]. Support for the PCH has been reported for face stimuli in fMRI studies showing larger RS when the probability of repetition is higher[9,10]. However, the evidence for a link between RS and prediction error is inconsistent in humans and absent in non-human primates (despite several confirmatory attempts)[11]. Neuronal fatigue-related accounts provide alternative explanations for RS, positing that RS results from input fatigue or inherited effects of upstream input areas[12]. Particularly for identical adapter (first presentation of a stimulus) and test stimuli (repeated stimulus presentation), RS at short-term timescales (in the order of deciseconds or seconds) may depend on the response magnitude to the adapter stimulus, which in turn may reflect response selectivity[13]. Of note, while there is evidence that RS does not depend on adapter response at the single neuron level[14,15], the population adapter response may determine RS, if RS depends on local network activity, as measured with fMRI.

Here, we use fMRI to test the PCH for RS for longer timespans in the context of mnemonic processing via differences due to memory for both short and long-term (48 h) visual memory for faces and houses in category-selective areas: Fusiform Face Area (FFA), posterior Superior Temporal Sulcus (pSTS), Parahippocampal Place Area (PPA) and RetroSplenial Cortex (RSC). Differences due to memory entail the predictive characteristic for future remembrance of neural signals during encoding and is assessed via the subsequent memory paradigm[16]. For our purpose, we adapted the subsequent memory paradigm and included a repetition during the encoding stage to investigate the interaction between RS and differences due to memory (Fig. 1).

According to predictive coding accounts of RS, events that are weakly encoded during the first presentation (adapter) will have a lower expectation rate at the second presentation and thus signal higher prediction error (i.e. reduced RS). Furthermore, these events will have a lower probability of future remembrance, compared to events that have been successfully encoded during the first encoding event, which will have a higher expectation rate and signal less prediction error (stronger RS) at the second encoding event (test stimulus) and have a higher probability of future remembrance[2]. While fatigue-related models have been formulated to explain short-term repetition effects, they are relevant for the present design as it has been suggested that long-term RS may relate to familiarity effects[17,18]. Long-term RS is likely to depend on these familiarity effects, which may be associated with stronger initial activation. On the other hand, they predict that RS will primarily depend on the activation level during the adapter presentation, particularly in the case of identical adapter-test stimuli, possibly reflecting response selectivity resulting from the proportion of responding cells. Furthermore, the amount of cells responding to an event will correlate with the probability of future remembrance[12]. Both PCH and fatigue accounts thus anticipate a positive interaction between RS and differences due to memory. We therefore first investigated which areas showed a compatible response pattern, i.e. an RS × differences due to memory interaction. However, the critical issue relates to the dependence of RS on the adapter response. Only fatigue accounts predict a positive association between RS and neural signal to the adapter. We therefore investigated absolute RS and relative RS (controlling for adapter response), and also correlated RS with response to the adapter stimulus. To account for regression to the mean and the mathematical coupling between baseline value and change, we used Oldham's method[19]. Oldham provided evidence that investigating the association between baseline measures and change from baseline can be performed by correlating the baseline value with the average of the initial and repetition value. Furthermore, we follow up on the association between encoding activation and recognition performance, i.e. differences due to memory, as PCH would predict a negative association between pooled activation over both presentations on the one hand (reflecting pooled prediction error), and recognition performance on the other hand.

Second, we investigated the predictive timeframe of the encoding activation as well as RS over longer lags. While most RS studies in the category-selective visual cortex focus on short-term effects in the order of seconds, we extend the timeframe to 10 min and 48 h. It is unlikely that local bottom-up effects in the category-selective cortex span an interval of minutes or days and hence any RS after longer lags is suggestive for familiarity mechanisms, presumably related to long-term plasticity, possibly via prefrontal and/or medial temporal memory-related inputs.

Fifty-six healthy participants were instructed to memorize 80 pictures of faces and 80 of houses. The pictures were semi-randomly presented one by one in the MR scanner and each picture was presented twice, i.e. once during each of two runs (Fig. 1a). Immediately after the encoding, participants performed an immediate recognition (IR, Fig. 1b) memory test. In addition, a delayed recognition (DR, Fig. 1c) test was performed in the scanner 48 h after the encoding. In each recognition test, all 160 stimuli from the encoding were intermixed with 80 new ones (40 houses and 40 faces) and presented one by one. Participants were instructed to indicate their remembrance and confidence of remembrance. Neural activation was estimated in category-selective areas for faces and houses in the visual cortex in each hemisphere defined at subject-level: FFA, posterior STS (pSTS), PPA, and RSC and this during 4-time points: first encoding, second encoding, IR, and DR. We used linear mixed model (LMM) analyses to investigate RS × recognition performance interactions, with and without accounting for response to the adapter.

Finally, previous research revealed evidence for stable individual differences in eye movements during face recognition. These eye movements play a functional role during face processing and how well people recognize faces[20,21]. We investigated fixation compliance and performed an eye-tracking experiment in an independent sample.

The present study reveals absolute, but not relative RS for lags up to 48 h. Absolute RS interacts with the probability of future remembrance in pSTS. This effect was not observed for relative RS, i.e. when controlled for adapter-response. Furthermore, the findings reveal an association between adapter response and RS, also after controlling for the mathematical dependence between both measures. These findings challenge predictive coding accounts of visual memory and are more compatible with adapter-related and familiarity accounts.

## Results

**V1 activation and eye movements.** There were no significant differences in V1 activation between the first and repeated encoding presentation, nor between immediate and delayed

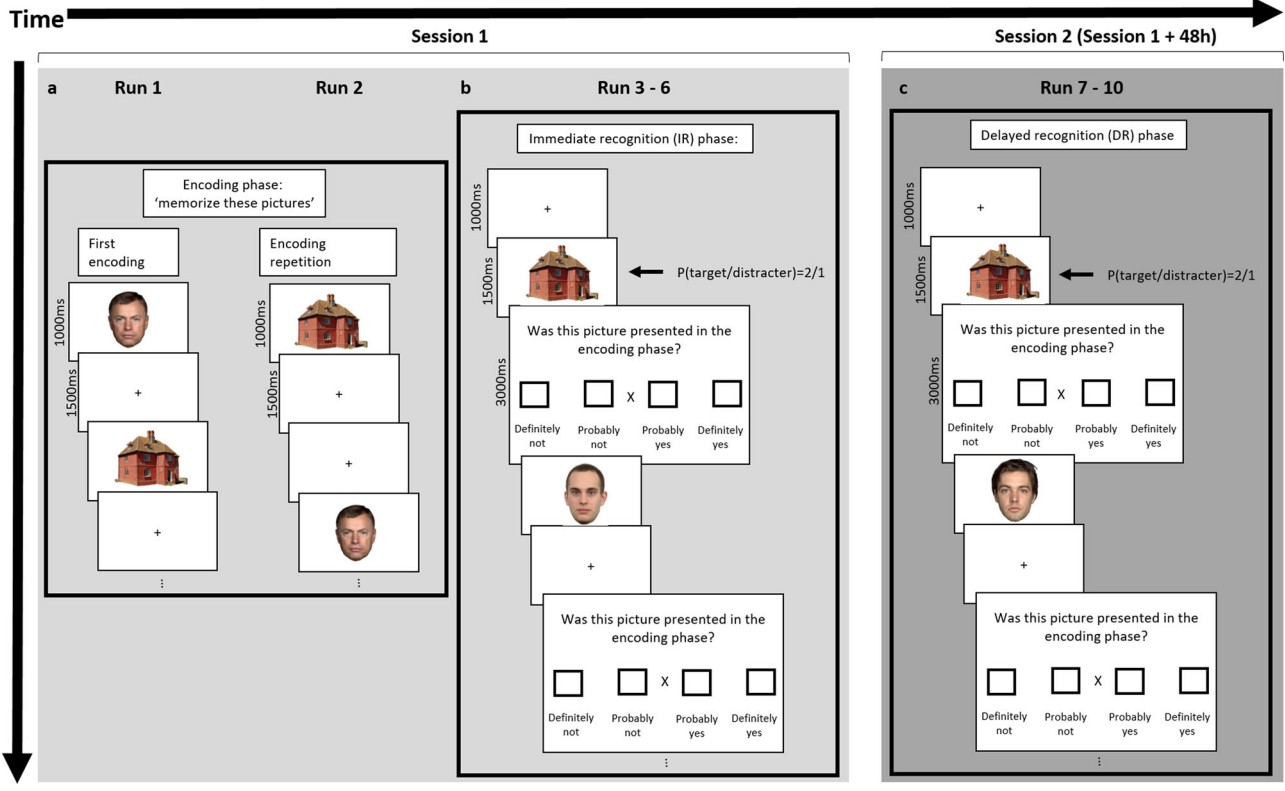

**Fig. 1 Schematic design of the protocol.** The experiment consisted of an encoding phase, an immediate recognition phase (IR), and a delayed recognition phase (DR), with a total of 10 runs. **a** The encoding phase consisted of two runs. In each run, 160 stimuli (80 houses and 80 faces) were pseudo-randomly presented for 1000 ms with a 1500 ms inter-stimulus interval. The second encoding run (encoding repetition) was identical to the first, except for the order of stimulus presentation. Prior to the encoding phase, participants were instructed to memorize the stimuli in order to recognize them in a subsequent recognition test. **b** The IR phase directly followed the encoding phase. In the IR phase, the stimuli consisted of the 160 stimuli from the encoding phase intermixed with an additional 80 distractors: 40 houses and 40 faces. The trials were equally divided over 4 runs during the IR phase. In the IR, the 1500 ms presentation of a stimulus was followed by a 3000 ms response screen, which displayed the question "Was this picture presented in the encoding phase?" with four boxes below referring to four response alternatives: "definitely not", "probably not", "probably yes", "definitely yes". **c** The DR phase was conducted two days after the first session (48 h). The procedure of the DR was identical to that of the IR, except for the stimulus presentation order and the distracter stimuli, i.e. 80 new distractors were presented in the DR.

recognition in V1, suggestive for similar retinal positions across timepoints and contra-indicating that fixation position-driven differences could explain RS in upstream areas.

Eye-movement results revealed task-compliant fixation on the fixation cross and similar fixation positions during the presentation of the face and house stimuli (See Supplementary Fig. 1). Furthermore, the pattern of fixation positions was similar between adapter and test. Mann Whitney U-tests revealed no significant difference in fixations between adapter and test for both categories (face, house, all $p$'s > 0.280).

**Behavioural results**. Mann–Whitney U-tests revealed no significant difference between d' for faces vs. houses for IR or DR (all $p$'s > 0.962, Fig. 2).

**RS during encoding as a function of immediate recognition peformance**. The mean lag between the adapter and the test stimulus was 9.59 min. A linear mixed model (LMM) analysis on the activation estimate with repetition (2 levels: adapter and test), immediate recognition performance (3 levels: forgotten, probably remembered or definitely remembered), repetition × immediate recognition performance as fixed effects and age, sex, and adapter-test time lag as confounding factors revealed a significant interaction in the right pSTS ($p = 0.014$, Bonferroni-corrected), with significant RS for definitely remembered stimuli ($p = 0.001$, Bonferroni-

corrected, Fig. 3) but not for probably remembered and forgotten stimuli (all $p$'s > 0.139).

In addition, LMM revealed increased RS for definitely remembered stimuli compared to forgotten stimuli ($p = 0.020$, Bonferroni-corrected). The effect reflected a 17% signal reduction for 'probably' (mean ± s.e.m = adapter: 0.672 ± 0.081, test: 0.558 ± 0.095) and 33% for 'definitely' (adapter: 0.735 ± 0.085, test: 0.496 ± 0.084) remembered stimuli compared to a 46% signal enhancement for 'forgotten' stimuli (adapter: 0.398 ± 0.110, test: 0.612 ± 0.109). The results thus also entail a qualitative effect, showing 'repetition enhancement' for events that will not be remembered versus RS for events that will be remembered. No interactions were observed in the other face regions (all $p$'s > 0.504, Fig. 3). In addition, we observed a main effect of repetition in the right FFA ($p = 0.029$).

The LMM on the relative RS index ((adapter-test)/adapter) abolished the interaction effect in the right pSTS ($p = 0.693$). The Pearson's correlation analysis between RS and the average of the adapter and test response revealed a significant result in the right pSTS ($r = 0.306$, $p = 0.026$, Fig. 4a). To account for outliers, we performed a control analysis in which we first calculated Z-scores and excluded participants with a Z-score higher than 3 and lower than −3 ($N = 1$). A similar Pearson's correlation analysis on the reduced sample again revealed a significant result (Z-score > 3, $r = 0.361$, $p = 0.008$).

For houses we observed no interactions in any of the regions (all $p$'s > 0.200, Fig. 3), in addition, we also did not observe a main effect of repetition (all $p$'s > 0.228) in any of the house regions.

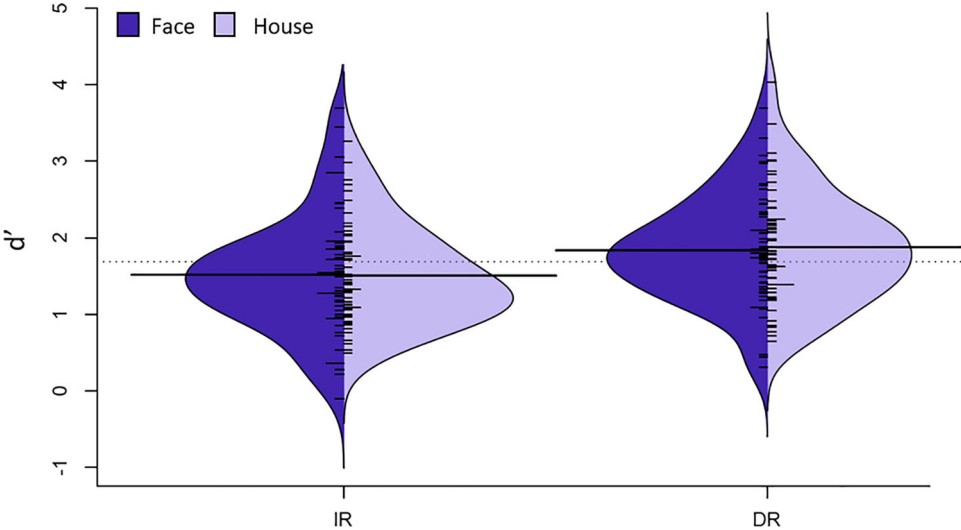

**Fig. 2 Behavioural results (d′) displayed by Split Violin plots.** Split Violin plots of d′ reveal no significant difference as a function of stimulus category (face vs. house) for IR or DR (all *p*'s > 0.962). Horizontal lines represent the mean. *N* = 54 healthy subjects.

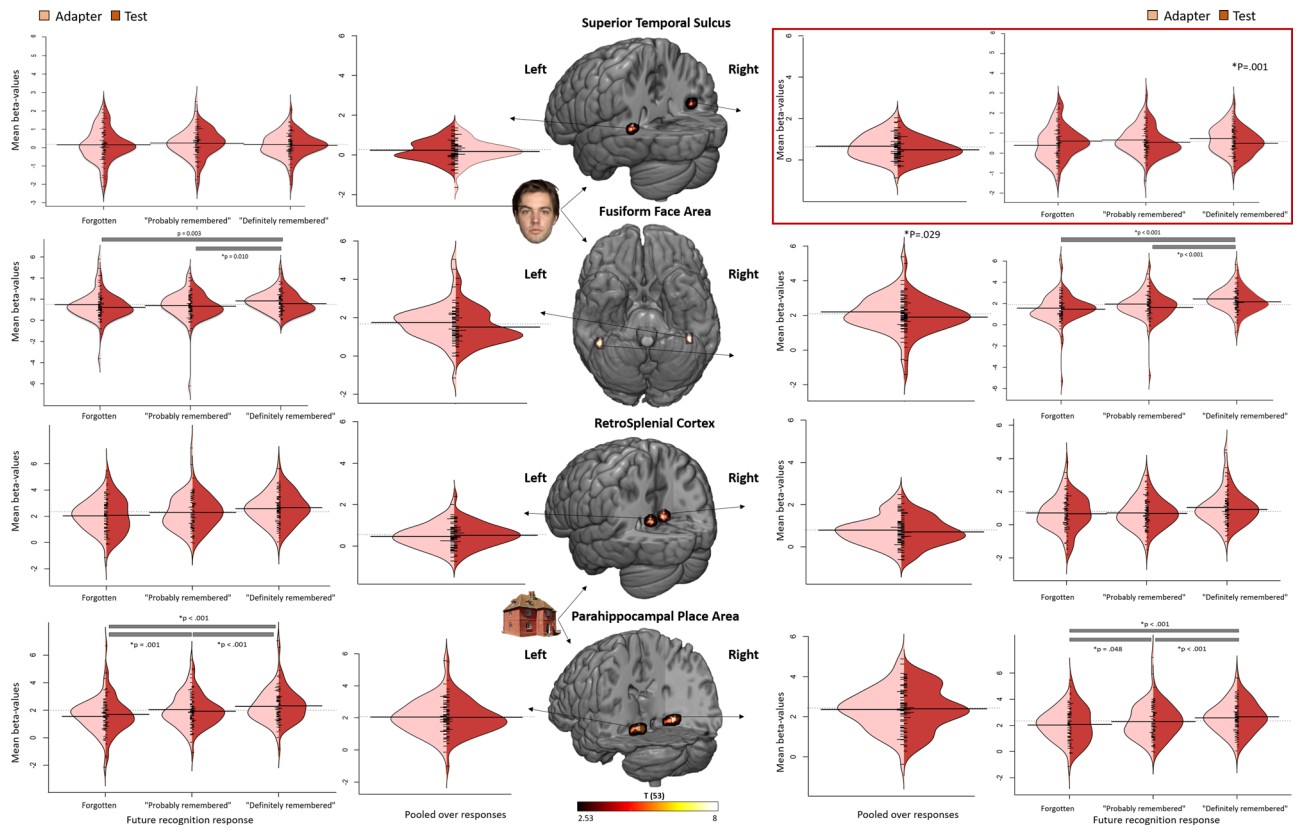

**Fig. 3 Interaction between RS and differences due to memory in right pSTS.** Split violin plots displaying effect size estimates (beta-values) as a function of ROI, repetition, and memory performance during the immediate recognition test. Horizontal bars indicate significant differences between performance levels. Linear mixed model analysis reveals a significant interaction between repetition suppression and differences due to memory in the right pSTS (*p* = 0.014, Bonferroni-corrected). The centre column displays the ROI locations and absolute *t*-values of the statistical maps (faces vs. houses and vice versa) on which the ROI definition is based (*q* < 0.005, FDR-corrected). Horizontal lines represent the mean. *N* = 54.

**RS during encoding as a function of delayed recognition performance**. A similar LMM but with delayed instead of immediate recognition performance revealed a significant repetition × delayed recognition performance interaction in the right pSTS (*p* = 0.035, Bonferroni-corrected), with significant RS for definitely remembered stimuli (*p* = 0.002, Bonferroni-corrected, Fig. 5), but not for probably remembered and

forgotten stimuli (all *p*'s > 0.384). The follow-up LMM revealed increased RS for the definitely remembered stimuli compared to forgotten stimuli (*p* = 0.041, Bonferroni-corrected). The effect reflected a 13% signal reduction for 'probably' (adapter: 0.620 ± 0.089, test: 0.538 ± 0.089) and 31% for 'definitely' (adapter: 0.721 ± 0.089, test: 0.494 ± 0.076) remembered stimuli compared to a 17% signal enhancement for 'forgotten' stimuli

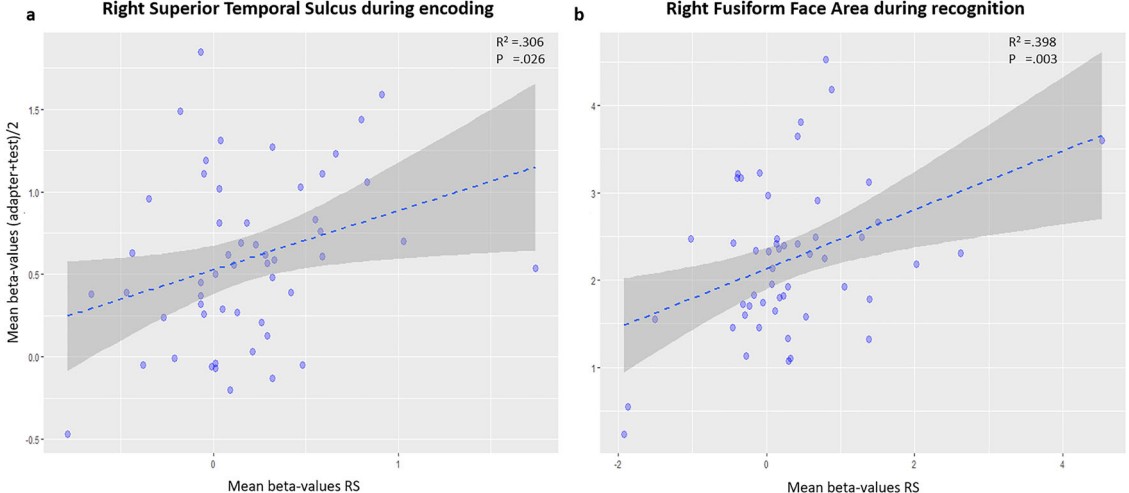

**Fig. 4 Scatterplot for the face-selective area. a** Two-sided Pearson's correlation using Oldham's method revealed a positive correlation ($p = 0.026$) between activation for initial response and RS in the right pSTS during encoding. **b** Two-sided Pearson's Correlation using Oldham's method revealed a positive correlation ($p = 0.003$) between activation during the initial response (immediate recognition; IR) and RS in the right FFA during recognition. $N = 54$.

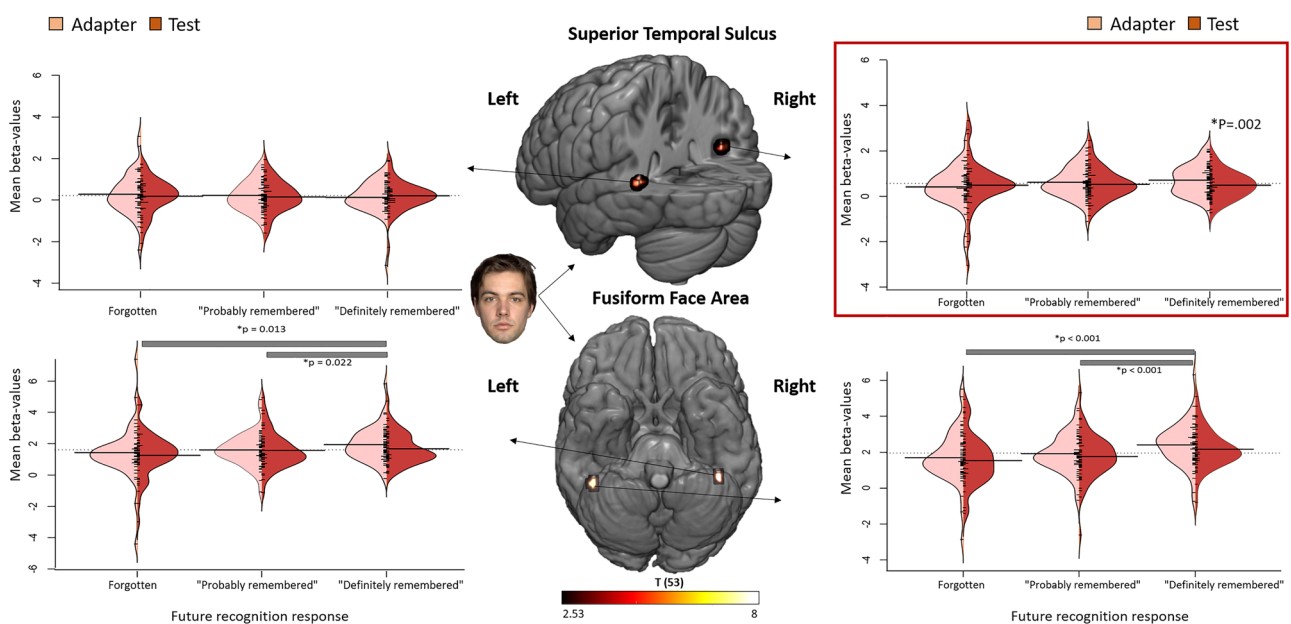

**Fig. 5 Interaction between RS and differences due to memory in right pSTS.** Split violin plots displaying effect size estimates (beta-values) as a function of ROI, repetition, and memory performance during delayed recognition test. Horizontal bars indicate significant differences between performance levels. Linear Mixed Model analysis reveals a significant interaction between repetition suppression and differences due to memory in the right STS ($p = 0.035$, Bonferroni-corrected). The centre column displays the ROI locations and absolute t-values of the statistical maps (faces vs. houses) on which the ROI definition is based ($q < 0.005$, FDR-corrected). Horizontal lines represent the mean. $N = 54$.

(adapter: $0.422 \pm 0.108$, test: $0.493 \pm 0.150$). No interactions were observed in the other face regions (all $p$'s > 0.277, Fig. 5).

The LMM on the relative RS index abolished the effect in the right pSTS ($p = 0.558$).

**RS during recognition**. We next investigated whether RS during recognition is associated with recognition performance by comparing activation during IR with DR. As the average event overlap between IR and DR was 34% for 'forgotten stimuli', 35% for 'probably remembered' stimuli, and 78% for 'definitely remembered' stimuli, we performed an LMM analysis on the activation estimate with repetition (2 levels: adapter (IR) and test (DR)) on the events from the performance condition 'definitely

remembered' only. Furthermore, the proportion event overlap was included as a covariate. This revealed a marginally significant result in the right FFA ($p = 0.050$).

Because of the partial event overlap in performance categories between IR and DR, we investigated RS pooled over performance levels. Therefore, we performed an LMM analysis on the activation estimate with repetition (2 levels: adapter (IR) and test (DR)). This revealed a significant effect in the right FFA ($p = 0.035$, Fig. 6). The effect reflected a 13% signal reduction (adapter: $2.389 \pm 0.158$, test: $2.078 \pm 0.114$). No significant effect was observed for the other regions (all $p$'s > 0.174).

As we investigated RS pooled over performance levels, we were not able to perform an LMM on the relative RS index.

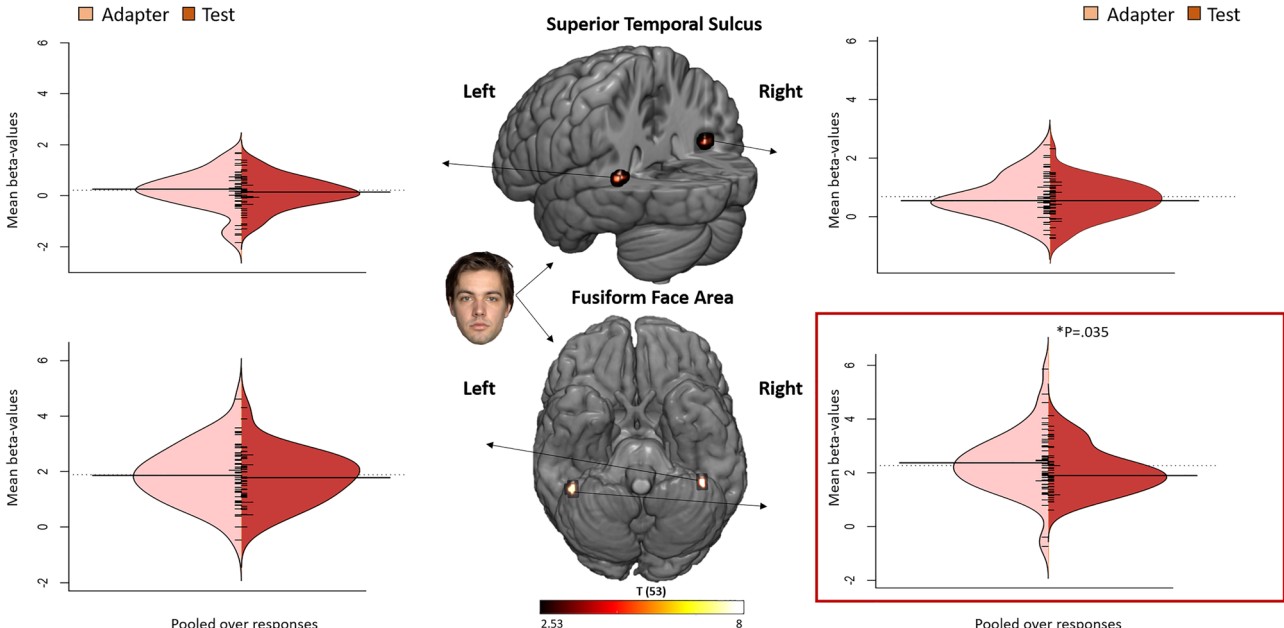

**Fig. 6 Main effect of repetition in the right FFA.** Split violin plots displaying effect size estimates (beta-values) as a function of ROI and repetition pooled over performance levels, during immediate (adapter) versus delayed (test) recognition task. This revealed a significant main effect of repetition in the right FFA ($p = 0.035$, Bonferroni-corrected). The centre column displays the ROI locations and absolute $t$-values of the statistical maps (faces vs. houses) on which the ROI definition is based ($q < 0.005$, FDR-corrected). Horizontal lines represent the mean. $N = 54$.

Pearson's correlation analysis between RS on the one hand and the average of the adapter and test, on the other hand, revealed a significant result in the right FFA ($r = 0.398$, $p = 0.003$; Fig. 4b). To account for outliers, we performed a control analysis in which we first calculated Z-scores and excluded participants with a Z-score higher than 3 and lower than −3 ($N = 1$). A similar Pearson's correlation analysis on the reduced sample again revealed a significant result (Z-score > 3, $r = 0.340$, $p = 0.015$).

**Pooled differences due to memory**. Subsequently, we investigated the difference due to memory for pooled activation over both encoding presentations. LMM on the encoding activation as a function of performance during immediate memory revealed a main effect of performance for the left and right FFA ($p < 0.001$). For houses, we observed a main effect of performance for the left and right PPA ($p < 0.001$), the right RSC ($p < 0.001$), and the left RSC ($p = 0.045$). LMM on the encoding activation as a function of performance during delayed memory revealed a main effect of performance for the left and right FFA ($p < 0.002$).

## Discussion

The PCH was addressed in the present study for RS during mnemonic processing in the category-selective visual cortex. While the typical adapter-test interval in RS fMRI studies lies in the order of seconds, the intervals in the present study extend these by factors of 200 and 86 k. Although RS typically reduces with increasing lags[22], our findings surprisingly reveal significant RS for lags of ~10 min and even of 48 h. A previous study did report long-term (3 days) RS in the object naming system, where subjects were instructed to name the real objects[23]. For the FFA, RS has been reported over 2.4 s intervals, but not over 225 s intervals[24]. The relatively higher power of the present study may explain this discrepancy and underlie one of the main novel findings, consisting of significant RS following a 48 h interval in the face-selective cortex. These long-term RS effects may be explained by neurophysiological mechanisms such as local cortical plasticity mechanisms akin to familiarity[25] and top-down

mnemonic mechanisms, presumably via interactions with the fronto-parietal working memory network and/or hippocampal memory network[26,27].

The main research question related to the compatibility between PCH and the interaction profile between RS and recognition memory performance. This was the case in the right pSTS for absolute RS, i.e. the net difference between the neural response to the adapter vs. the test stimulus. The results during encoding revealed a positive association with memory performance, in line with PCH. This fits with the notion of a reduced prediction error for strongly encoded events[2]. The findings showed an analogous qualitative pattern with signal suppression for subsequently remembered events and signal enhancement for subsequently forgotten events. Yet, the RS-memory performance interaction is also compatible with alternative hypotheses, positing that RS relates to the adapter response. According to adapter response-related accounts, a stronger adapter response may cause both more RS and stronger familiarity effects[17,18], as reflected in the typical difference due to memory effects[28]. We therefore investigated relative RS in which we controlled for the adapter response and observed that the significant RS × memory performance interactions disappeared. Furthermore, after controlling for the mathematical dependence between adapter response and RS, our findings reveal a moderate yet significant correlation between RS and adapter response in the right pSTS for the 10 min lag and the right FFA for the 48 h adapter-test lag, inconsistent with PCH. The results thus strongly adhere to adapter-related accounts. Additional conflicting findings with the PCH consisted of the typical differences due to memory for both the 10 min and 48 h lag in the left and right FFA, reflecting a positive association between activation and performance, contrary to PCH predictions[2], which would predict a negative association between activation and performance.

We did not observe an interaction between RS and performance in the FFA, and could as such not extend the findings of Summerfield et al.[10], reporting larger RS when the probability of repetition is higher in FFA. This may relate to methodological characteristics such as the lag, number of intervening events, and

task. The mean lag for faces in our study is 573 s with 160 inter-repetition events on average, while Summerfield et al. used a lag of 500 ms without inter-repetition events[10]. Repetition without intervening events allows for the formation of detailed expectations about the next stimulus, while this was task-irrelevant in the present study. Furthermore, previous studies have mainly investigated repetition probability by comparing RS for blocks with a high probability of encountering repetition versus blocks with a low probability of repetition[9,10]. A study examining the interaction between face processing and predictive coding observed that activity in the pSTS is associated with the level of familiarity of faces in the environment[29]. In addition, previous research found that STS reactivation and structural hippocampus integrity predicted subsequent recognition performance[30]. These findings are in line with the idea of long-term RS depending on familiarity effects. On the other hand, the study of Apps and Tsakiris suggested that FFA activity is associated with the amount of familiarity updating of a face[29]. This may indicate that activity in the FFA reflects the prediction error, which updates the familiarity of the face stimuli[29], and therefore reveals a decline in activity each time the same face is presented. These findings could suggest an association between RS and difference due to memory for the STS on the one hand and an association between RS and the amount of familiarity updating in the FFA on the other hand.

We found a category-dependent RS effect, observing RS in the right FFA and interaction between RS and difference due to memory in the right pSTS, but in none of the house selective areas. Previous studies reported RS for houses in the PPA[31]. We hypothesise that any inconsistencies on RS in PPA may relate to methodological issues, such as the lag and number of intervening events. The mean lag in our study is 587 s with 160 inter-repetition events on average, while Weiner et al. used a maximum lag of 174 s with 87 intervening events[32]. The preference in the medial ventrotemporal cortex for house stimuli predicts RS for houses. However, this effect disappeared when the lag was increased from .5 s to a mean lag of 20 s[32]. These findings suggest different longevity of RS between categories, with long-term RS in the face-selective cortex, but not in the house-selective cortex.

Previous research revealed that individuals show distinctive eye movement strategies when identifying faces and these strategies are stable over time[20]. Individual differences in fixation position are less likely to account for RS effects, as RS in higher-order areas is less susceptible to changes in retinal positions[33]. In a recent study, stimulus-selective adaptation was measured while presenting adapter and test stimuli at different positions. The results revealed that stimulus-selective adaptation occurred for distances up to 18°[33]. Furthermore, neurons in the inferior temporal cortex have larger receptive fields compared to earlier visual areas[34,36]. These combined findings suggest that it is unlikely that the observed RS effects in higher-order areas are driven by differences in fixation. A limitation of the present study is that we only focussed on category-selective areas, however, the mechanism underlying RS may be region dependent[32], and support for mnemonic prediction error signals has been reported in hippocampal functional connectivity and activity patterns[26,37]. The hippocampus may facilitate both prediction and memory, by inhibiting neocortical prediction errors[37].

Furthermore, the focus of this paper was on RS in higher-level visual areas, rather than on inter-individual fixation differences in encoding strategies. Therefore, given the present results, it would be relevant to investigate any associations with fixation position and participant characteristics in future studies.

The present study reveals absolute, but not relative RS for lags up to 48 h. Absolute, but not relative RS over a 10 min lag positively interacts with the probability of future remembrance in pSTS. Furthermore, the findings reveal an association between adapter response and RS, also after controlling for the mathematical dependence between both measures. We consider these combined findings as challenging for predictive coding models of visual memory. The findings seem more compatible with the alternative hypotheses positing adapter-related accounts, suggesting that long-term RS may relate to familiarity effects.

## Methods
The study was approved by the Ethical Committee of University Hospitals Leuven. All subjects gave written informed consent in accordance with the Declaration of Helsinki.

**Participants**. All subjects had normal or corrected-to-normal visual acuity. Fifty-six subjects participated in our study. They were recruited via advertisements for participation in an fMRI memory experiment. One participant was excluded due to technical failure during fMRI acquisition and one participant was excluded due to an indication of pathology. The final sample for the main analyses thus consisted of fifty-four subjects [13 males (24%), mean age ± SD = 34 ± 11 years, range 21–64].

**Stimuli**. The stimuli consisted of 160 pictures of houses, which were selected from our own database and stripped of visual background. 80 (40 female) neutral and 80 (40 female) angry faces were selected from our database and other face stimulus databases[38,39] and stripped of visual background. Participants gave consent for the display of their faces during the experiment and for the creation of Figures. All stimuli were resized to 400 pixels in height. The size of the stimuli was between 9° and 13° vertically and 9° and 13° horizontally of visual angle. For faces, the distance between the eyes and mouth was approximately 4° degrees of visual angle. For the current study, we only analysed category effects (face and house).

**Procedure**. The experiment consisted of an encoding phase, an immediate recognition (IR) phase, and a delayed recognition phase (DR). The encoding phase consisted of two blocks, each in a separate run of 8.48 min. In each run, 160 stimuli (80 houses and 80 faces) were pseudo-randomly presented for 1500 ms against a white background, separated by a 1000 ms ISI during which a black fixation cross was presented. In addition, 40 null-events (1500 ms) were included during which only the fixation cross was displayed. The second encoding run was identical to the first, except for the order of stimulus presentation. Prior to the encoding phase, participants were instructed to memorize the stimuli of the encoding phase in order to recognize them in a subsequent recognition test.

The IR phase directly followed the encoding phase. In the IR phase, the stimuli consisted of the 160 stimuli from the encoding phase intermixed with an additional 80 distractors: 40 houses and 40 faces. Sixty null-events were interspersed. The procedure in the IR phase consisted of the presentation of a stimulus (1500 ms) followed by a response screen (3000 ms). The response screen displayed the question "Was this picture presented in the encoding phase?" with four boxes below referring to four response alternatives: "definitely not", "probably not", "probably yes", "definitely yes". In the centre of the response screen, and "X" was presented. Participants were instructed to move the "X" to the left or right by pressing the corresponding button on a 2-button response box. A fixation screen (500 ms) followed the response screen, after which the next trial started. The 300 trials (160 targets + 80 distracters + 60 null-events) were equally divided over 4 runs of 7.48 min each and each comprising 75 trials: 30 houses (20 from the encoding phase), 30 faces (20 from the encoding phase), and 15 null-events. In none of the encoding or recognition blocks, there were more than three consecutive stimuli of the same category (face, house, or null-event).

The DR is conducted two days after the first session. The DR is identical to the IR, except for the stimulus presentation order and the distracter stimuli, i.e. 80 new distractors were presented in the DR in order to minimize source-recognition difficulties.

The recognition phase began with five practice trials with car stimuli, which were included to familiarize the participants with the response procedure. See Fig. 1 for a schematic design of the procedure. Pictures were projected onto a screen and were viewed through a mirror mounted on the head coil, minimizing head movements. Responses were recorded via an MR-compatible keypad (MRI Devices, Waukesha, WI), positioned on the right side of the participant's abdomen. A desktop workstation running PRESENTATION® 19.0 (Neurobehavioral Systems, San Francisco, CA) controlled stimulus presentation and response registration.

**Eye movements**. To investigate differences in the retinal positions of the stimuli between adapter and test we conducted an additional eye tracker experiment outside the scanner using the same procedures. This was performed in an independent sample of 20 participants that were demographically matched to the fMRI sample at the group level [6 males (30%), mean age ± SD = 37 ± 20 years, range 21–67]. Independent Sample $t$-test showed that no significant differences were detected for age ($p = 0.538$). The $X^2$ test showed no significant differences for sex ($p = 0.604$).

Eye movement data were collected at a sampling rate of 120 Hz using the Tobii eye tracker TX300[40] and Tobii Studio 3.4.7. A five-point fixation position calibration was performed prior to the experiment. We applied default settings, including the Tobii fixation filter, with a velocity threshold of 0,84 pixels/ms (35 pixels) and a distant threshold (distance between two consecutive fixations) of 35 pixels (default). A detailed description of the Tobii fixation can be found in the Tobii Studio user manual (https://www.tobiipro.com/siteassets/tobii-pro/user-manuals/tobii-pro-studio-user-manual.pdf). Before data acquisition, we created three Areas of Interest (AOI), corresponding to the mouth, nose, and eyes within Tobii Studio. Subsequently, we created group-level heatmaps during the presentation of three conditions: fixation, faces, and houses, using default eye tracker settings (https://www.tobiipro.com/siteassets/tobii-pro/user-manuals/tobii-pro-studio-user-manual.pdf). The full eye tracker experiment is described in detail in the Supplementary method

**V1 activation**. In order to estimate differences in eye movements, we performed an additional analysis of the imaging data, to control if there were any systematic differences in the V1 representation of the stimuli between the first versus repeated encoding presentation and between immediate versus delayed recognition. In order to account for Type II errors, we applied a more liberal threshold ($P_{height} < 0.01$, uncorrected, minimal cluster threshold of 12 voxels).

**Statistical analyses**. Behavioural results were analysed according to signal detection theory[41]. R-Score Plus[42] was used to calculate d' for confidence rating designs. D' was calculated as a function of the category (face vs. house). Parametric testing depended on the results of a Shapiro-Wilk test. We calculated the mean interval between stimulus repetitions (lag) for every participant as a function of the stimulus category.

**Brain imaging**. Brain imaging was performed on a 3 T Siemens Achieva scanner, using a 32-channel head coil. Acquisition parameters for 45 participants consisted of a high-resolution T1-weighted anatomical image (voxel size: $0.98 \times 0.98 \times 1.20$ mm$^3$) using a 3D turbo field echo sequence (TR:9.6 ms, TE:4.6 ms, matrix size:256 × 256, 182 slices), a T2*-weighted GE-EPI sequence with the following parameters: TR: 2000 ms; TE, 30 ms, matrix size: 80 × 78, FOV: 230 mm, flip angle: 90°, slice thickness: 4 mm, no gap, axial slices: 38. For the other 9 participants, a similar high-resolution T1-weighted anatomical image was acquired (voxel size: $1.10 \times 1.10 \times 1.10$ mm$^3$) using a 3D turbo field echo sequence (TR:6.9 ms, TE:3.2 ms, matrix size:256 × 256, 208 slices) and a T2*-weighted GE-EPI sequence with the following parameters: TR: 2000 ms, TE: 30 ms, matrix size: 80 × 78, FOV: 230 mm, flip angle: 90°, slice thickness: 4 mm, no gap, axial slices: 36. Scan acquisition setting was included as a nuisance variable in all brain imaging analyses.

**Brain imaging analysis**. Imaging data were analyzed using BrainVoyager 21.4[43]. Pre-processing of functional data consisted of slice scan time correction, temporal high-pass filtering to remove low-frequency drifts, realignment to the first image to compensate for head motion, and spatial smoothing with a Gaussian filter of 4 mm FWHM. Functional data were co-registered with the anatomical images and normalized into Talairach coordinate space.

At the first level, the statistical analysis was based on the general linear model (GLM) with repetition (no, yes), category (face, house), and subsequent memory performance (not, probably yes, and definitely yes) as factors. The 'definitely not' and 'probably not' conditions were pooled as 25 participants did not use both categories during their experiment in one of both categories. Null-events were modelled explicitly. At a second level, a random-effects GLM was performed.

Only ROI analyses were performed in category-selective areas: Fusiform Face Area (FFA), posterior Superior Temporal Sulcus (pSTS), Parahippocampal Place Area (PPA), and RetroSplenial Cortex (RSC). These were identified by contrasting all distractor face trials with all distractor house trials ($q < 0.005$, FDR-corrected). This revealed significant clusters comprising the topography of bilateral FFA and pSTS (faces > houses) on the one hand and PPA and RSC (houses > faces) on the other hand. Each of the 8 ROIs was subsequently defined at the subject level by creating a 3 mm-radius sphere around the subject-specific peak-voxel. The statistical threshold for ROI-definition at the subject level was set at $P_{height} < 0.005$. Two participants did not show any significant results for the right FFA, five for the left FFA, one for the right pSTS, two for the left pSTS, and four for the left RSC. These proportions adhere to previous studies[44–46]. These participants were excluded from the respective analyses. Subsequently, we used the trials in which target stimuli were presented to perform the ROI-analyses. Beta-values were extracted for each subject for faces and houses for each encoding run separately as a function of performance on the subsequent memory test.

To evaluate the anticipated outcomes for the three different memory stages (encoding, IR, and DR), we performed Linear Mixed Model analyses, for every ROI separately on the data from the preferred stimulus category. First, we investigated the RS × differences due to memory interaction for absolute RS. We identified 3 levels of performance: 'forgotten', 'probably remembered', and 'definitely remembered'. The 'forgotten' level consisted of the pooling of the 'probably not' and 'definitely not' response categories.

Linear Mixed Models (LMM) were estimated with the beta-values as a dependent variable with repetition (2 levels: adapter and test), performance (3 levels: forgotten, probably remembered, or definitely remembered), and repetition × performance as fixed effects using an unstructured variance-covariance matrix (based on a Likelihood test). We applied Bonferroni-correction to account for multiple comparisons. Follow-up interactions were focused on within-performance category differences. In addition to the main effects of performance and repetition, we performed post hoc analyses to study the interaction between performance and region and the interaction between repetition and region.

Second, we investigated the timespan of the mnemonic prediction error signal. We performed a similar LMM, but with future performance based on DR instead of IR.

Third, we investigated whether the RS signal during recognition is associated with recognition performance by comparing activation during IR with DR, by performing a similar LMM.

To follow up on significant repetition × performance interactions, we performed a LMM analyses on RS, defined as the beta-value during the first presentation (adapter) minus the beta-value during the second presentation (test). LMM was estimated with the beta-values as a dependent variable and performance as a fixed effect using an unstructured variance–covariance matrix (based on a Likelihood test). We applied Bonferroni-correction for multiple comparisons. All LMMs controlled for age, sex, and adapter test time lag (i.e. included these covariates as fixed effects in the model).

In order to investigate relative RS, we repeated all analyses but controlled for adapter response (i.e. (adapter test)/adapter) to calculate the RS index.

In addition, to investigate the interdependence between adapter and RS we performed a two-sided Pearson's correlation using Oldham's method[19]. In order to investigate whether outliers drive any significant correlation, we performed an additional two-sided Pearson's correlation analysis excluding outliers with a Z-score greater than 3 or less than −3.

**Statistics and reproducibility**. Imaging data were analyzed using BrainVoyager 21.4[43]. All statistical analyses were performed on the whole participant group ($n = 54$). Statistical analyses were performed using SPSS[47]. We used linear mixed model (LMM) analyses to investigate RS × recognition performance interactions, with and without accounting for response to the adapter. LMM with Bonferroni's multiple comparisons post hoc tests were employed with $P$ values < 0.05 considered significant. To investigate the interdependence between adapter and RS we performed a two-sided Pearson's correlation using Oldham's method[19]. Values are expressed as the mean ± s.e.m., except for the sample size (mean ± SD).

**Reporting summary**. Further information on research design is available in the Nature Research Reporting Summary linked to this article.

## Data availability
The data supporting the findings of this study are available within the paper and its Supplementary Information. Source data are provided as Supplementary Data 1.

## Code availability
All data sets (including a Source Data file) are published in a publicly available Dryad digital repository (https://doi.org/10.5061/dryad.x95x69pj3).

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

## Acknowledgements

J.V.D.S. is supported by a KU Leuven Starting Grant + C2 + Sequoia Fund. We wish to thank Wim Van der Elst for his contribution to the statistical procedures.

## Author contributions

D.S., S.S., R.P., C.S., L.V., L.E., M.V., and J.V.d.S. contributed to the conception and design of the study. D.S. and J.V.d.S. organized the database. D.S., R.V., M.V., and J.V.d.S. interpreted the results. D.S., Y.H., and K.V. performed the statistical analysis. D.S. wrote the first draft of the manuscript. All authors contributed to manuscript revision, read and approved the submitted version.

## Competing interests

The authors declare no competing interests.
