## [Transparent Peer Review File · Communications Biology]

Reviewers' comments:

Reviewer #1 (Remarks to the Author):

Stam and colleagues examine the relationship between repetition suppression (RS) and visual memory processing in category-selective areas in both typical participants and those with developmental prosopagnosia while considering theoretical predictions of the predictive coding hypothesis. The authors claim that the findings support a predictive coding account of RS, but there are several major methodological flaws along the way that prevent any sound theoretical conclusions at this time. That being said, the paper is not suitable for *Communications Biology* or a more specialized journal until the authors re-analyze their data while considering several circularity issues. Major and minor concerns are detailed below.

Major Concerns:

1. There are at least two tiers of circularity in the analyses presently conducted by the authors. First, the same data are used to define the regions of interest (ROIs) as well as to conduct the main series of analyses.

In terms of ROI definition, the authors use the same data to first define the ROIs and then perform their subsequent analyses. Thus, the authors cannot ignore biases that are introduced in the analyses given this circular approach. A likely fix is to first use a subset of the data to define the ROIs and then, to use the rest of the data to examine their original experimental question.

Second, the authors perform an analysis in which they examine the correlation between RS and fMRI responses to the adapter stimulus (the first presentation). Even though the authors describe this analysis as "a more stringent test we also correlated RS with response to the adapter stimulus" (Pg. 4, lines 80-81), this is far from a stringent test. Specifically, RS is determined based on the comparison between the response to the adapter image and the repeated image. Therefore, the RS response and the response to the adapter image are not independent and these analyses are circular.

Given these circularity issues, it is complicated to assess the theoretical predictions proposed by the authors when considering the present analyses. For both concerns, the authors should be clear about how independence of selection and test data was handled. Presently, independence was not considered at all.

2. Two additional levels of concern that affect ROI definitions and then, theoretical implications of potential results. First, the extensive amount of spatial smoothing. 8mm of spatial smoothing is extensive given modern methods and norms for modern datasets (for example, the Human Connectome Project).

Second, the authors report a series of statistical thresholds to identify their circular ROIs on page 19 in which they use one (very) liberal threshold to identify ROIs at the group level ($p < .05$) and then another ($p < .005$) at the individual subject level. In addition to their circular approach, the effort to define these regions brings doubt to the quality of their individual subject data considering the fact that face- and place-selective regions are some of the most reproducibly identified areas across labs (Kanwisher, 2010 PNAS). These methodological issues may explain why the data from the present authors are at odds with previously published RS data. For example, previously published studies show that the PPA shows significant RS for repeated presentations of either places/scenes (Epstein et al., 2008 J Neurophysiol; Weiner et al., 2010 J Neurophysiol) or even faces (Mur et al., 2010 Cereb Cortex). These data are at odds with the present study.

3. The developmental prosopagnosia (DP) component seems like a complete afterthought to the manuscript and not fully integrated into the experimental question. Specifically, the present framing of the paper hinges on predictive coding hypotheses associated with repetition suppression (RS) and

visual memory processing in category-selective areas. As it stands, the DP component is not integrated and I think the authors should consider if a correlation with 11 participants (Fig. 7) is meaningful or not (especially since they have 56 "typical" participants).

4. The data presentation could be significantly improved. For example, the caption to Fig. 1 is as follows: "Schematic design presentation." The authors should not assume that the reader has the internal information that the authors have to immediately understand the specifics of the procedure from the images in the Figure. It is common for the result depicted in the Figure to be reflected in the title of the caption. The authors, instead, highlight the method used in the title of each of the figures and the captions do not explain what the novel finding is that we are supposed to see, which brings to question: What do the authors want the reader to see in each of these figures?

Despite the circularity issues and other issues flagged above, you do have a rich dataset.

Minor concerns:

1. While the authors stress that previous studies do not consider the long term nature of RS, a previous study (uncited by the present authors) published in 2005 showed that RS effects last over a period of 3 days (van Turennout et al. 2005 Nat Neuroscience).

2. In the Introduction, the authors write: "Furthermore, it has been postulated that the reduced neural response following repeated presentation of a stimulus, termed repetition suppression (RS) reflects a reduction in prediction error and as such a facilitation of a behavioral response, e.g. faster reaction time during identification." While this interpretation is appealing, the authors should also consider that previous work emphasizes that there is often a lack of correspondence between RS and behavioural priming (summarized in Grill-Spector, 2008).

3. In Fig. 2, there is a clear change in the distribution for DP scores for faces between IR and DR. This difference is never referenced. Why might the distribution change between the IR and DR conditions for DPs?

Reviewer #2 (Remarks to the Author):

Title

Long term fMRI adaptation depends on adapter response in face-selective cortex

Overall Assessment

This paper uses fMRI and repetition suppression (RS) to test the Predictive Coding Hypothesis (PCH) for memory and recall of unfamiliar faces and houses in both healthy controls (HC) and subjects with developmental prosopagnosia (DP). The authors find that the strength of the initial encoding (the "adapter") predicted RS after both 10 minute and even 48 hour delay periods. Support for the PCH in STS arises from a relationship between the absolute size of RS between study and test in the STS. However, this relationship was found to arise from a relationship between the size of the initial response to the adapter and subsequent magnitude of RS (and recognition memory). Overall, the study was well-designed with clear questions, hypotheses, and predictions. My main issue is with fixation (eye movements), and the apparent lack of strict control on the retinotopic position of the retinal stimulus image. If the authors can provide the information and data requested below, or provide a compelling argument for why that data does not need to be included, I have no problem approving this well-written paper for publication. The findings, especially of evidence of RS after such long delay times, are novel and important for the field. Thank you to the authors for good work, look forward to hearing your replies!

Issues to address

1) I would like to see data regarding sizes in visual angle (retinal image sizes)

a. What were the sizes of the stimuli in degrees visual angle?

- Total vertical and horizontal size?
 - For faces, what was the distance between the eyes and mouth?
- b. Where was the fixation cross positioned relative to the stimuli?
- For faces in particular, where was the fixation cross positioned relative to the eyes and mouth?
- 2) I get the impression that fixation at the fixation cross was instructed but not enforced and/or monitored during the fMRI scans. Is that the case?
- a. This is critical due to the stable individual differences in where people tend to look on faces, and corresponding differences in how well people can recognize faces depending on where they look.
- i.e., Peterson & Eckstein, *Psych Science*, 2013 and Stacchi et al., *Journal of Neuroscience*, 2019. See also Peterson et al., *Journal of Vision*, 2019 for relevant potential group differences between neurotypical subjects, NTs, and subjects with developmental prosopagnosia, DPs.
- b. The concern here is if subjects were faithfully fixating the cross, then subjects who tend to look close to that common location would be expected to have a better retinotopic match between the retinal stimulation and their (retinotopically tuned) internal representations.
- This is especially important for any comparisons between groups, such as NTs vs DPs where one group may be given an uncontrolled and unmeasured variation in a hidden experimental parameter: the mismatch between where they are requested to fixate and where they prefer to fixate.
- c. If, on the other hand, subjects did not faithfully hold fixation at the cross, and instead gravitated their gaze to their preferred fixation location, then at least subjects were all equated in terms of their retinotopic matching to their internal representations.
- Though that leaves inter-subject differences in the retinal positions of the other stimuli, houses, uncontrolled and unmeasured).
- d. In either case, I think it's critical to have a measure of where each subject was actually fixating on each image to attribute any RS or behavioral differences to a precise mechanism (e.g., strength of initial adapter encoding predicts subsequent RS and delayed recognition performance due to mismatches in fixation and retinotopic encoding).
- e. Finally, claims regarding the maintenance of RS over long delay periods (like 48 hours here) require measurement of fixation position at the two measurement times. If, for whatever reason, some subjects fixated differently in the two sessions, then that would be expected to drive neural response differences by itself.
- 3) Should probably add some conservative language regarding the DP findings. They are interesting and thought-provoking for sure, but should be couched in the disclaiming context of a small sample size (n = 11).

Reviewers' comments:

Reviewer #1 (Remarks to the Author):

Reviewer #1	Stam and colleagues examine the relationship between repetition suppression (RS) and visual memory processing in category-selective areas in both typical participants and those with developmental prosopagnosia while considering theoretical predictions of the predictive coding hypothesis. The authors claim that the findings support a predictive coding account of RS, but there are several major methodological flaws along the way that prevent any sound theoretical conclusions at this time. That being said, the paper is not suitable for Communications Biology or a more specialized journal until the authors re-analyze their data while considering several circularity issues. Major and minor concerns are detailed below.
Reply	We thank Reviewer #1 for the assessment of the manuscript. We like to point out that we do not interpret the findings as unambiguous support for a predictive coding account of RS. Only for absolute RS, but not for relative RS, the findings are in line with a predictive coding account. Furthermore, the absolute RS findings must be further nuanced by the observation of an association between the response to the adapter and RS (also following the revised analyses). We consider these combined findings as challenging for predictive coding models of visual memory and have clarified this in the revised manuscript: “The present study reveals absolute, but not relative RS for lags up to 48h. Absolute, but not relative RS over a 10 minute lag positively interacts with probability of future remembrance in pSTS. Furthermore, the findings reveal an association between adapter response and RS, also after controlling for the mathematical dependence between both measures. We consider these combined findings as challenging for predictive coding models of visual memory. The findings seem more compatible with the alternative hypotheses positing adapter-related accounts, suggesting that long-term RS may relate to familiarity effects.” (Pg. 15-16, lines 367-373).

Major Concerns:

Reviewer #1	1. There are at least two tiers of circularity in the analyses presently conducted by the authors. First, the same data are used to define the regions of interest (ROIs) as well as to conduct the main series of analyses. In terms of ROI definition, the authors use the same data to first define the ROIs and then perform their subsequent analyses. Thus, the authors cannot ignore biases that are introduced in the analyses given this circular approach. A likely fix is to first use a subset of the data to define the ROIs and then, to use the rest of the data to examine their original experimental question.
Reply	We thank Reviewer #1 for providing this helpful suggestion and have re-analysed the data accordingly. We have used independent data to define the ROIs on the one hand and perform the ROI-analysis on the other hand. In particular, we have used the trials in which distracter stimuli were presented to define the category selective areas. Subsequently, we used the trials in which target stimuli were presented to perform the ROI-analyses. We have described the revised procedures in the methods section: “Only ROI analyses were performed in category selective areas: Fusiform Face Area (FFA), posterior Superior Temporal Sulcus (pSTS), Parahippocampal Place Area (PPA), and RetroSplenial Cortex (RSC). These were identified by contrasting all distracter face trials with all distracter house trials ($q < .005$, FDR-corrected). This revealed significant clusters comprising the topography of bilateral FFA and pSTS (faces > houses) on the one hand and PPA and RSC

(houses>faces) on the other hand. Each of the 8 ROIs was subsequently defined at subject level by creating a 3mm-radius sphere around the subject-specific peak-voxel. The statistical threshold for ROI-definition at subject level was set at $P_{height} < .005$. Two participants did not show any significant results for the right FFA, five for the left FFA, one for the right pSTS, two for the left pSTS, and four for the left RSC. These proportions adhere to previous studies (Kanwisher et al., 1997; Pitcher et al., 2019; Bilalić et al., 2019). These participants were excluded from the respective analyses. Subsequently, we used the trials in which target stimuli were presented to perform the ROI-analyses. Beta-values were extracted for each subject for faces and houses for each encoding run separately as a function of performance on the subsequent memory test.” (Pg. 22-23, lines 518-531).

Reviewer #1 Second, the authors perform an analysis in which they examine the correlation between RS and fMRI responses to the adapter stimulus (the first presentation). Even though the authors describe this analysis as "a more stringent test we also correlated RS with response to the adapter stimulus" (Pg. 4, lines 80-81), this is far from a stringent test. Specifically, RS is determined based on the comparison between the response to the adapter image and the repeated image. Therefore, the RS response and the response to the adapter image are not independent and these analyses are circular.

Reply **We are grateful to Reviewer #1 for pointing out this issue. We have addressed this by revising the methods and included optimized procedures to address this research question. In particular, we used Oldham’s method, which was proposed as a means to address exactly this circularity. Oldham (Oldham 1962) provided evidence that investigating the association between baseline measures and change from baseline can be performed by correlating the change and the average of the initial and repetition value. We have described the revised procedures in the methods section:**

“To account for regression to the mean and the mathematical coupling between baseline value and change, we used Oldham’s method (Oldham, 1962). Oldham provided evidence that investigating the association between baseline measures and change from baseline can be performed by correlating the baseline value with the average of the initial and repetition value.” (Pg. 4-5, lines 87-91).

Reviewer #1 Given these circularity issues, it is complicated to assess the theoretical predictions proposed by the authors when considering the present analyses. For both concerns, the authors should be clear about how independence of selection and test data was handled. Presently, independence was not considered at all.

Reply **As outlined in the replies above, we have addressed the rightful concerns of Reviewer #1 regarding circularity by means of re-analyses of the data. We have revised the methods by using procedures that account for circularity in ROI definition and ROI analyses on the one hand by using independent datasets and for initial value and change on the other hand by using a tailored correlational approach that accounts for the raised issue.**

Reviewer #1	2. Two additional levels of concern that affect ROI definitions and then, theoretical implications of potential results. First, the extensive amount of spatial smoothing. 8mm of spatial smoothing is extensive given modern methods and norms for modern datasets (for example, the Human Connectome Project).
Reply	We have followed up on the suggestion of Reviewer #1 and re-analysed the data with only half of the spatial smoothing amount of the initial analysis (i.e. 4mm vs 8mm). “Pre-processing of functional data consisted of slice scan time correction, temporal high-pass filtering to remove low-frequency drifts, realignment to the first image to compensate for head motion, and spatial smoothing with a Gaussian filter of 4mm FWHM.” (Pg. 22, lines 508-512). Using the revised methods (procedures that account for circularity in ROI definition and half of the spatial smoothing) only slightly changed our results. The main change in our findings was that LMM analysis on the revised data revealed increased RS for definitely remembered stimuli compared to forgotten stimuli, but not for probably remembered stimuli compared to forgotten stimuli, as we observed with the previous data. The main conclusions of the paper were not substantially affected by the re-analyses. The rewritten result section can be found on pages 6-9, lines 127-210.

New figures, representing the re-analysed data:

Fig. 3 Interaction between RS and differences due to memory in the right pSTS. Split violin plots displaying effect size estimates (beta-values) as a function of ROI, repetition, and memory performance during immediate recognition test. Horizontal bars indicate significant difference between performance levels. Linear Mixed Model analysis reveals a significant interaction between repetition suppression and differences due to memory in the right pSTS ($p=.014$, Bonferroni-corrected). The centre column displays the ROI locations and absolute t-values of the statistical maps (faces vs houses) on which the ROI definition is based ($q<.005$, FDR-corrected). $N=54$ healthy subjects.

Fig. 4 Interaction between RS and differences due to memory in right pSTS. Split violin plots displaying effect size estimates (beta-values) as a function of ROI, repetition, and memory performance during delayed recognition test. Horizontal bars indicate significant difference between performance levels. Linear Mixed Model analysis reveals a significant interaction between repetition suppression and differences due to memory in the right STS ($p=.035$, Bonferroni-corrected). The centre column displays the ROI locations and absolute t-values of the statistical maps (faces vs houses) on which the ROI definition is based ($q<.005$, FDR-corrected). $N=54$ healthy subjects.

Fig. 5 Scatterplot for the face-selective area. **a** Two-sided Pearson's correlation using Oldham's method revealed a positive correlation ($p=.026$) between activation for initial response and RS in the right pSTS during encoding. **b** Two-sided Pearson's correlation using Oldham's method revealed a positive correlation ($p=.007$) between activation during initial response (immediate recognition; IR) and RS in the right FFA during recognition. $N=54$ healthy subjects.

Fig. 6 Main effect of repetition in the right FFA. Split violin plots displaying effect size estimates (beta-values) as a function of ROI and repetition pooled over performance levels, during immediate (adapter) versus delayed (test) recognition task. This revealed a significant main effect of repetition in the right FFA ($p = .035$; Bonferroni-corrected). The centre column displays the ROI locations and absolute t-values of the statistical maps (faces vs houses) on which the ROI definition is based ($q < .005$, FDR-corrected). $N = 54$ healthy subjects.

Reviewer #1	Second, the authors report a series of statistical thresholds to identify their circular ROIs on page 19 in which they use one (very) liberal threshold to identify ROIs at the group level ($p < .05$) and then another ($p < .005$) at the individual subject level. In addition to their circular approach, the effort to define these regions brings doubt to the quality of their individual subject data considering the fact that face- and place-selective regions are some of the most reproducibly identified areas across labs (Kanwisher, 2010 PNAS).
Reply	We thank Reviewer #1 for this remark. However, it may be based on a partly accurate conception of the initial method, in which the ROI definition at group level was performed using a very conservative correction procedure for multiple comparisons, i.e. the Bonferroni procedure. In the revised manuscript, we have identified category selective areas via an initial whole-brain group analyses in which we contrast faces with houses on the distractor trials using a statistical threshold of $q < .005$, FDR-corrected. To identify the ROIs at subject level, the statistical threshold was set at $P_{\text{height}} < .005$. If no neural category-effect was found in a specific ROI, the subject was excluded for the analyses of this specific ROI, as reported above. We acknowledge the high reproducibility of these regions and also observe this in the present dataset, which consists of an event-related design, as opposed to higher-powered blocked-designs which are typically used to define these regions.

Reviewer #1	These methodological issues may explain why the data from the present authors are at odds with previously published RS data. For example, previously published studies show that the PPA shows significant RS for repeated presentations of either places/scenes (Epstein et al., 2008 J Neurophysiol; Weiner et al., 2010 J Neurophysiol) or even faces (Mur et al., 2010 Cereb Cortex). These data are at odds with the present study.
Reply	We agree with Reviewer #1 that our data shows inconsistency with previously published RS data in the PPA. We hypothesise that any inconsistencies on RS in PPA may relate to methodological differences such as the lag between repetitions and number of intervening events. The mean lag in our study is 587 s with 160 inter-repetition events on average, while Weiner et al. used a maximum lag of 174 s with 87 intervening events (Weiner et al., 2010). In addition, the RS effect for houses in the medial ventrotemporal cortex disappeared when the lag was increased from .5 s to a mean lag of 20 s (Weiner et al., 2010). Furthermore, Epstein et al. (2008) measured RS as the difference between new and repeated stimuli, while in the current study we assessed RS by the difference in activation for the same stimulus presented as adapter and test. On the other hand, the present study has a higher power compared with previous research due to the large sample size, extending the sample size of other studies by a factor 3 to 7 (our study: n=56; Epstein et al., 2008: n=16; Weiner et al., 2010: n=9; Mur et al., 2010: n=8). Hence, we hope to have provided convincing evidence here and above to document that inconsistencies with previous studies may have other causes than poor data quality in the present study.

Reviewer #1	3. The developmental prosopagnosia (DP) component seems like a complete afterthought to the manuscript and not fully integrated into the experimental question. Specifically, the present framing of the paper hinges on predictive coding hypotheses associated with repetition suppression (RS) and visual memory processing in category-selective areas. As it stands, the DP component is not integrated and I think the authors should consider if a correlation with 11 participants (Fig. 7) is meaningful or not (especially since they have 56 "typical" participants).
Reply	We thank Reviewer #1 for raising this issue. The rationale for including DP was related to the cognitive modality that we focus on in the study, i.e. face memory. As a deficit in this ability is the core feature of prosopagnosia, we considered this a relevant condition to critically test any hypotheses that emerged from the findings in healthy control subjects and provided an opportunity to report converging evidence. However, as Reviewer#1's comment implies, this reasoning is less evident and thus we have followed-up on the remark and omitted the prosopagnosia section in the revised manuscript.

Reviewer #1	4. The data presentation could be significantly improved. For example, the caption to Fig. 1 is as follows: "Schematic design presentation." The authors should not assume that the reader has the internal information that the authors have to immediately understand the specifics of the procedure from the images in the Figure. It is common for the result depicted in the Figure to be reflected in the title of the caption. The authors, instead, highlight the method used in the title of each of the figures and the captions do not explain what the novel finding is that we are supposed to see, which brings to question: What do the authors want the reader to see in each of these figures?
Reply	Thank Reviewer #1 for pointing this out. We modified Figure 1 to clarify the design presentation. In addition, we modified the title and the captions of the figures to clarify.

Fig. 1 Schematic design of the protocol. The experiment consisted of an encoding phase, an immediate recognition phase (DR), and a delayed recognition phase (DR), with a total of 10 runs. The encoding phase (left panel) consisted of two runs. In each run, 160 stimuli (80 houses and 80 faces) were pseudo-randomly presented. The second encoding run (encoding repetition) was identical to the first, except for the order of stimulus presentation. Prior to the encoding phase, participants were instructed to memorize the stimuli of the encoding phase in order to recognize them in a subsequent recognition test. The IR phase (middle panel) directly followed the encoding phase. In the IR phase, the stimuli consisted of the 160 stimuli from the encoding phase intermixed with an additional 80 distractors: 40 houses and 40 faces. The trials were equally divided over 4 runs during the IR phase. In the IR, presentation of a stimulus was followed by a response screen, which displayed the question “Was this picture presented in the encoding phase?” with four boxes below referring to four response alternatives: “definitely not”; “probably not”; “probably yes”; “definitely yes”. The DR (right panel) was conducted two days after the first session (48h). The procedure of the DR was identical to that of the IR, except for the stimulus presentation order and the distracter stimuli, i.e. 80 new distractors were presented in the DR.

Reviewer #1	Despite the circularity issues and other issues flagged above, you do have a rich dataset.
Reply	We thank Reviewer #1 for praising our dataset.

Minor concerns:

Reviewer #1	1. While the authors stress that previous studies do not consider the long term nature of RS, a previous study (uncited by the present authors) published in 2005 showed that RS effects last over a period of 3 days (van Turennout et al. 2005 Nat Neuroscience).
Reply	We thank Reviewer #1 for pointing this out. We updated the text and referred to the study of Van Turennout et al. 2000 in the revised manuscript: "A previous study did report long-term (3 days) RS in the object naming system, where subjects were instructed to name the real objects (Van Turennout et al., 2000)." (Pg. 11, lines 265-266).

Reviewer #1	2. In the Introduction, the authors write: "Furthermore, it has been postulated that the reduced neural response following repeated presentation of a stimulus, termed repetition suppression (RS) reflects a reduction in prediction error and as such a facilitation of a behavioral response, e.g. faster reaction time during identification." While this interpretation is appealing, the authors should also consider that previous work emphasizes that there is often a lack of correspondence between RS and behavioural priming (summarized in Grill Spector, 2008).
Reply	We thank Reviewer #1 for pointing this out. We updated the introduction accordingly, including a reference to the respective study, supplemented with references to other studies that report similar findings. "Furthermore, it has been postulated that the reduced neural response following repeated presentation of a stimulus, termed repetition suppression (RS) reflects a reduction in prediction error and as such a facilitation of a behavioral response (priming), e.g. faster reaction time during identification. While some studies suggest a quantitative relation between priming and RS (Maccotta & Buckner, 2004; Zago et al., 2005) others suggest that RS may not directly relate to visual priming (Grill-Spector, 2008; Sayres & Grill-Spector, 2006; McMahan & Olson, 2007)." (Pg. 3, lines 42-46).

Reviewer #1	3. In Fig. 2, there is a clear change in the distribution for DP scores for faces between IR and DR. This difference is never referenced. Why might the distribution change between the IR and DR conditions for DPs?
Reply:	We thank Reviewer #1 for pointing this out. We hypothesize that this may reflect different phenotypic presentations of prosopagnosia, i.e. apperceptive and amnesic prosopagnosia. However, as reported above to the previous comments of Reviewer#1, we have omitted the prosopagnosia section from the revised manuscript.

Fig. 2 Behavioral results (d') displayed by Split Violin plots. Split Violin plots of d' reveal no significant difference as a function of stimulus category (face vs house) for IR or DR (all p 's > .962). N=54 healthy subjects.

Reviewer #2 (Remarks to the Author):

Title

Long term fMRI adaptation depends on adapter response in face-selective cortex

Overall Assessment

This paper uses fMRI and repetition suppression (RS) to test the Predictive Coding Hypothesis (PCH) for memory and recall of unfamiliar faces and houses in both healthy controls (HC) and subjects with developmental prosopagnosia (DP). The authors find that the strength of the initial encoding (the “adapter”) predicted RS after both 10 minute and even 48 hour delay periods. Support for the PCH in STS arises from a relationship between the absolute size of RS between study and test in the STS. However, this relationship was found to arise from a relationship between the size of the initial response to the adapter and subsequent magnitude of RS (and recognition memory). Overall, the study was well-designed with clear questions, hypotheses, and predictions. My main issue is with fixation (eye movements), and the apparent lack of strict control on the retinotopic position of the retinal stimulus image. If the authors can provide the information and data requested below, or provide a compelling argument for why that data does not need to be included, I have no problem approving this well-written paper for publication. The findings, especially of evidence of RS after such long delay times, are novel and important for the field. Thank you to the authors for good work, look forward to hearing your replies!

Issues to address

Reviewer #2	1) I would like to see data regarding sizes in visual angle (retinal image sizes) a. What were the sizes of the stimuli in degrees visual angle? a. Total vertical and horizontal size?
	Reply: We thank Reviewer #2 for pointing out this missing information. The size of the stimuli was between 9° and 13 ° vertically and between 9 ° and 13° horizontally of visual angle. We have clarified this in the revised manuscript (Pg. 17, line 398-399).

Reviewer #2	a.2. For faces, what was the distance between the eyes and mouth?
Reply	We thank Reviewer #2 for pointing out this missing information. For faces, the distance between the eyes and mouth was approximately 4° degrees of visual angle. We have clarified this in the revised manuscript (Pg. 17-18, line 399-400).

Reviewer #2	b. Where was the fixation cross positioned relative to the stimuli? For faces in particular, where was the fixation cross positioned relative to the eyes and mouth?
Reply	We thank Reviewer #2 again for pointing out this missing information. For the face stimuli, the fixation cross was positioned around the midpoint of the virtual line connecting the nasion and nasal septum. This implies that the relative position of the fixation cross was closer to the eyes than to the mouth. We added this information to the supplementary materials and added an illustrative figure in the supplementary materials (Fig. S1a). The red fixation cross illustrates where the fixation cross was positioned relative to the stimuli.

Fig S1a: **Position fixation cross.** Red fixation cross illustrates where the fixation cross was positioned relative to the stimuli. For the face stimuli the fixation cross was positioned around the midpoint of the virtual line connecting the nasion and nasal septum.

Reviewer #2	2) I get the impression that fixation at the fixation cross was instructed but not enforced and/or monitored during the fMRI scans. Is that the case? a. This is critical due to the stable individual differences in where people tend to look on faces, and corresponding differences in how well people can recognize faces depending on where they look.  • i.e., Peterson & Eckstein, Psych Science, 2013 and Stacchi et al., Journal of Neuroscience, 2019. See also Peterson et al., Journal of Vision, 2019 for relevant potential group differences between neurotypical subjects, NTs, and subjects with developmental prosopagnosia, DPs.
Reply	Reviewer#2 is correct that fixation at the fixation cross was instructed but not enforced. In the present study, subjects were instructed to fixate at the fixation point, but we did not measure eye movements and relied on their compliance to the instruction. However, we added a control study using eyetracking to address this. We have clarified this in the revised manuscript: “Finally, previous research revealed evidence for stable individual differences in eye movements during face recognition. These eye movements play a functional role during face processing and how well people recognize faces (Peterson and Eckstein, 2013; Stacchi et al., 2019). We investigated fixation compliance and performed an eye-tracking experiment in an independent sample.” (Pg. 6, lines 122-125).

Reviewer #2	b. The concern here is if subjects were faithfully fixating the cross, then subjects who tend to look close to that common location would be expected to have a better retinotopic match between the retinal stimulation and their (retinotopically tuned) internal representations. This is especially important for any comparisons between groups, such as NTs vs DPs where one group may be given an uncontrolled and unmeasured variation in a hidden experimental parameter: the mismatch between where they are requested to fixate and where they prefer to fixate.
Reply	We thank Reviewer #2 for pointing this out. Based on the comments of Reviewer #1, we have omitted the prosopagnosia section from the revised manuscript, which makes the present remark of Reviewer#2 less relevant for the revised manuscript.
Reviewer #2	c. If, on the other hand, subjects did not faithfully hold fixation at the cross, and instead gravitated their gaze to their preferred fixation location, then at least subjects were all equated in terms of their retinotopic matching to their internal representations. Though that leaves inter-subject differences in the retinal positions of the other stimuli, houses, uncontrolled and unmeasured).
Reply:	We thank Reviewer #2 for this remark. We have followed up on this comment and included it in the discussion paragraph focussing on suggestions for future studies. We like to point out that the focus in the present study was however not on inter-individual fixation differences in encoding strategies, but rather on RS in higher level visual areas. Previous research indicated that repetition suppression in higher visual areas like STS and FFA are at least partly tolerant to changes in retinal position. Neurons in inferior temporal cortex have larger receptive fields compared to earlier visual areas (Kobatake & Tanaka, 1994; Desimone & Gross, 1979; Op De Beeck & Vogels, 2000). In addition, RS in higher order areas is also less susceptible to changes in retinal positions. In a recent study, stimulus-selective adaptation was measured while presenting adapter and test stimuli at different positions. The results revealed that stimulus-selective adaptation occurred for distances up to 18° (Fabbrini & Vogels, 2020). The comment of Reviewer#2 is therefore particularly relevant to extend in depth on the present results and investigate any associations with fixation position and participant characteristics. We have included this consideration in the revised manuscript: “Furthermore, the focus of this paper was on RS in higher level visual areas, rather than on inter-individual fixation differences in encoding strategies. Therefore, given the present results, it would be relevant to investigate any associations with fixation position and participant characteristics in future studies.” (Pg. 15, lines 361-364).

Reviewer #2	d. In either case, I think it's critical to have a measure of where each subject was actually fixating on each image to attribute any RS or behavioral differences to a precise mechanism (e.g., strength of initial adapter encoding predicts subsequent RS and delayed recognition performance due to mismatches in fixation and retinotopic encoding).
Reply	We agree with Reviewer#2 that it is important to monitor whether participants indeed fixated at the fixation cross. To follow up on the remark of Reviewer#2, we conducted an additional eye tracker experiment outside the scanner using the same procedures (fixation at the fixation cross was instructed but

not enforced). This was performed in an independent sample of 20 participants that were demographically matched to the fMRI sample at group level [6 males (30 %); mean age \pm SD = 37 \pm 20 years, range 21-67]. Independent Sample t-test showed that no significant differences were detected for age ($p=.538$). Chi-Square test showed no significant differences for sex ($p=.604$).

Eye movement data was collected at a sampling rate of 120 Hz using the Tobii eye tracker TX300 (Tobii Technology AB, Sweden) and Tobii Studio 3.4.7. A five-point fixation position calibration was performed prior to the experiment. We applied default settings, including the Tobii fixation filter, with a velocity threshold of 0,84 pixels/ms (35 pixels) and a distant threshold (distance between two consecutive fixations) of 35 pixels (default). A detailed description of the Tobii fixation can be found in the Tobii Studio user manual

(<https://www.tobii.com/siteassets/tobii-pro/user-manuals/tobii-pro-studio-user-manual.pdf>). Before data acquisition, we created 3 Areas of Interest (AOI), corresponding to the mouth, nose and eyes within Tobii Studio.

Subsequently, we created group level heatmaps during presentation of 3 conditions: fixation, faces, and houses, using default eye tracker settings

(<https://www.tobii.com/siteassets/tobii-pro/user-manuals/tobii-pro-studio-user-manual.pdf>).

The results show that during the presentation of the fixation cross, the vast majority of fixations cover the area where the fixation cross was shown for all participants (Fig. S1b and S1c). The fixation cross was positioned around the center of the stimuli, which for faces consisted of the midpoint of the virtual line connecting the nasion and nasal septum. Overall, the results reveal task-compliant fixation on the fixation cross and a majority of similar fixation positions during presentation of the face and house stimuli.

Finally, we analysed the difference in number of fixations between the adapter and test stimuli, within each heatmap. Mann Whitney U test revealed no significant difference in fixations over time for both categories (face, house; all $p's > .280$).

We added the above information as a supplementary material section to the revised manuscript.

Fig. S1: Position and heatmap fixation cross. **a.** Red fixation cross illustrates where the fixation cross was positioned relative to the stimuli. For the face stimuli the fixation cross was positioned around the midpoint of the virtual line connecting the nasion and nasal septum. **b.** Group level (n=20) heatmaps of adapter stimuli, during presentation of 3 conditions: fixation, faces and houses. **c.** Group level (n=20) heatmaps of test stimuli, during presentation of 3 conditions: fixation, faces and houses.

Reviewer #2	e. Finally, claims regarding the maintenance of RS over long delay periods (like 48 hours here) require measurement of fixation position at the two measurement times. If, for whatever reason, some subjects fixated differently in the two sessions, then that would be expected to drive neural response differences by itself.
Reply	We followed up on the remark of Reviewer#2 and investigated differences in neural response at the longer delay (48h), as well as at the shorter delay. As any differences in fixation position would primarily and most prominently be reflected in topographic activity changes in primary visual cortex (V1), we compared activation in V1 during the first vs repeated encoding presentation and during immediate versus delayed recognition. In order to account for Type II errors, we applied a more liberal threshold ($P_{\text{height}} < .01$, uncorrected, minimal cluster threshold of 12 voxels). The results revealed no systematic differences over the 48h nor 10 minute delay in V1 representations, suggesting comparable fixation positions at the two measurement times and contra-indicating that fixation position-driven differences could explain RS in upstream areas. Furthermore, as Reviewer#2 reported above, previous research revealed that individuals show distinctive eye movement strategies when identifying faces and these strategies are stable over time (Peterson and Eckstein, 2013). In addition, we replied above that individual differences in fixation position are less likely to account for RS effects, as previous research revealed evidence that RS in higher order areas is less susceptible changes in retinal positions (Fabbrini & Vogels, 2020). Therefore, we hypothesise that individual and inter-subject differences in fixation over time will have no major influence on RS when studying RS in higher order areas. We have clarified this in the revised manuscript: “4.2.1.2 V1 activation and eye movements In order to estimate differences in eye movements, we performed an additional analysis of the imaging data, to control if there were any systematic differences in the V1 representation of the stimuli between the first versus repeated encoding presentation and between immediate

	versus delayed recognition. In order to account for Type II errors, we applied a more liberal threshold ($P_{\text{height}} < .01$, uncorrected, minimal cluster threshold of 12 voxels). To investigate differences in the retinal positions of the stimuli between adapter and test we conducted an additional eye tracker experiment outside the scanner using the same procedures. The eye tracker experiment is described in detail in the supplementary materials.” (Pg. 19, lines 435-443). “2.1 V1 activation and Eye movements The V1 activation analysis did not reveal any significant differences between the first and repeated encoding presentation, nor between immediate and delayed recognition in V1. Eye-movement results revealed task-compliant fixation on the fixation cross and similar fixation positions during presentation of the face and house stimuli. Furthermore, the pattern of fixation positions was similar between adapter and test. Mann Whitney U tests revealed no significant difference in fixations between adapter and test for both categories (face, house; all p's > .280).” (Pg. 6, lines 129-137). “The results revealed no systematic differences over the 48h nor 10 minute delay in V1 representations, suggesting comparable fixation positions at the two measurement times and contra-indicating that fixation position-driven differences could explain RS in upstream areas. Furthermore, previous research revealed that individuals show distinctive eye movement strategies when identifying faces and these strategies are stable over time (Peterson and Eckstein, 2013). In addition, individual differences in fixation position are less likely to account for RS effects, as previous research revealed evidence that RS in higher order areas is less susceptible changes in retinal positions (Fabbrini & Vogels, 2020). In a recent study, stimulus-selective adaptation was measured while presenting adapter and test stimuli at different positions. The results revealed that stimulus-selective adaptation occurred for distances up to 18° (Fabbrini & Vogels, 2020). In addition, previous research observed that neurons in inferior temporal cortex have larger receptive fields compared to earlier visual areas (Kobatake & Tanaka, 1994; Desimone & Gross, 1979; Op De Beeck & Vogels, 2000). These combined findings suggest that it is unlikely that the observed RS effects in higher order areas are driven by differences in fixation.” (Pg. 12, lines 274-285).
--	--

Reviewer #2	3) Should probably add some conservative language regarding the DP findings. They are interesting and thought-provoking for sure, but should be couched in the disclaiming context of a small sample size (n = 11).
Reply:	We thank Reviewer #2 for raising this issue. In line with the comment of Reviewer#1, we have omitted the prosopagnosia section from the revised manuscript.

REVIEWERS' COMMENTS:

Reviewer #1 (Remarks to the Author):

The authors have sufficiently addressed each of my previous concerns and I only have one remaining minor comment. Specifically, the authors should test whether or not outliers are driving the correlations reported in Figure 5.

Reviewers' comments:

Reviewer #1 (Remarks to the Author):

Reviewer #1	The authors have sufficiently addressed each of my previous concerns and I only have one remaining minor comment. Specifically, the authors should test whether or not outliers are driving the correlations reported in Figure 5.
Reply	We thank Reviewer #1 for this remark. To account for outliers, we performed a control analysis in which we first calculated Z-scores and excluded participants with a Z-score higher than 3 and lower than -3. In addition, we performed a similar Pearson's correlation analysis on the reduced sample. Both correlations remained significant (Pg. 7, lines 153-156; pg. 9 lines 190-194). We have described the additional analysis in the methods section: "In addition, to investigate the interdependence between adapter and RS we performed a two-sided Pearson's correlation using Oldham's method. In order to investigate whether outliers drive any significant correlation, we performed an additional two-sided Pearson's correlation analysis excluding outliers with a z-score greater than 3 or less than -3. (Pg. 19-20, lines 453-456).